# Risk Allele Frequency Analysis of Single-Nucleotide Polymorphisms for Vitamin D Concentrations in Different Ethnic Group

**DOI:** 10.3390/genes12101530

**Published:** 2021-09-28

**Authors:** Byung-Woo Yoon, Hyun-Tae Shin, Jehyun Seo

**Affiliations:** 1Division of Oncology, Department of Internal Medicine, Inje University Seoul Paik Hospital, Seoul 04551, Korea; bwyoon.md@gmail.com; 2Department of Dermatology, Inha University School of Medicine, Incheon 22212, Korea; hyuntae.shin@inha.ac.kr; 3Veterans Medical Research Institute, Veterans Health Service Medical Center, Seoul 05368, Korea

**Keywords:** Vitamin D deficiency, global prevalence, allele frequency, single nucleotide polymorphism, East-Asians, latitude

## Abstract

The prevalence of vitamin D deficiency varies from 20.8% to 61.6% among populations of different ethnicities, suggesting the existence of a genetic component. The purpose of this study was to provide insights into the genetic causes of vitamin D concentration differences among individuals of diverse ancestry. We collected 320 single-nucleotide polymorphisms (SNPs) associated with vitamin D concentrations from a genome-wide association studies catalog. Their population-level allele frequencies were derived based on the 1000 Genomes Project and Korean Reference Genome Database. We used Fisher’s exact tests to assess the significance of the enrichment or depletion of the effect allele at a given SNP in the database. In addition, we calculated the SNP-based genetic risk score (GRS) and performed correlation analysis with vitamin D concentration that included latitude. European, American, and South Asian populations showed similar heatmap patterns, whereas African, East Asian, and Korean populations had distinct ones. The GRS calculated from allele frequencies of vitamin D concentration was highest among Europeans, followed by East Asians and Africans. In addition, the difference in vitamin D concentration was highly correlated with genetic factors rather than latitude effects.

## 1. Introduction

Vitamin D, a fat-soluble vitamin, plays an essential role in bone mineralization and calcium homeostasis. Its deficiency is closely related to metabolic bone disease [1] and non-skeletal conditions, such as cardiovascular, infectious, and autoimmune diseases, as well as malignancies and diabetes [2,3,4,5]. Vitamin D is produced in the skin from 7-dehydrocholesterol by UV irradiation. Serum 25-hydroxy-vitamin D (25(OH)D_3_), the major circulating biomarker of vitamin D status, is converted to active vitamin D, 1,25(OH)_2_D, primarily in the kidney and, to a lesser extent, in the extra-renal tissue [6]. Serum vitamin D levels are strongly influenced by numerous factors, including age, obesity, skin color, dietary intake, exposure to ultraviolet B (UVB) sunlight, geographical latitude, and dietary supplements [7]. Studies have estimated that 1 billion people worldwide have vitamin D deficiency or insufficiency [1,8] which is a significant public health concern [7]. 

According to a global overview, the prevalence of vitamin D deficiency or the average vitamin D concentrations varies according to ethnicity. For instance, despite the serum 25(OH)D_3_ cutoff point being set at 20 ng/mL for adults, 54% of patients with African ancestry (the average 25(OH)D_3_ concentrations: 21.0 ± 10.4 ng/mL) fall below this level, compared to only 18% of Europeans ancestry (29.2 ± 10.9 ng/mL) in the United States [9]. In addition, deficiency status even differed at similar latitudes, as it ranged from 18.6% (26.0 ± 7.04 ng/mL) among Norwegians to 60.3% (18.0 ng/mL) among the Finnish [9]. With regard to East Asians, 32.1–53.9% of adults (20.4–23.4 ng/mL) in China and 53.6% in Japan (22.4 ± 7.5 ng/mL) have 25(OH)D_3_ concentrations of ≤20 ng/mL [9]. The deficiency prevalence or the average 25(OH)D_3_ concentrations is similar in Korea, and according to the Korea National Health and Nutrition Examination Survey, 47.3% of males (21.2 ± 7.5 ng/mL) and 64.5% of females (18.2 ± 7.1 ng/mL) have this deficiency [10].

The average 25(OH)D_3_ concentration differences among ethnic groups point to a possible genetic component. In recent years, genome-wide association studies (GWASs) have revealed that several significant loci, including *GC*, *NADSYN1*/*DHCR7*, *CYP2R1*, and *CYP24A1,* are associated with 25(OH)D_3_ deficiency [11,12,13,14,15,16,17]. These loci have been reported to function in the metabolism of vitamin D by converting into active form in the skin, liver, and kidney [2,13,14,15,16,17]. Previous researchers have suggested using whole-genome sequencing data of healthy subjects to identify disease phenotypes [18]. By combining data from the GWAS catalog of the National Human Genome Research Institute-European Bioinformatics Institute (NHGRI-EBI) and data from whole-genome sequencing of healthy subjects, it may be possible to identify risk-modeling of single-nucleotide polymorphisms (SNPs) related to 25(OH)D_3_ levels. Our study group has previously applied SNP-related models to two ophthalmic diseases and published significant research results [19,20]. There is, however, a need to identify the risk models based on genetic and real-world data of 25(OH)D_3_ levels and from populations of different ethnic groups to analyze these associations. This study aimed to identify the genetic causes and allele frequency differences related to 25(OH)D_3_ concentration among populations of diverse ancestry. Moreover, it aimed to compare the composite genetic risk scores using SNPs related to 25(OH)D_3_ concentrations for different ethnic groups.

## 2. Materials and Methods

### 2.1. Ethical Considerations

This study was approved by the Institutional Review Board (IRB) of the Veterans Health Service Medical Center, Korea (IRB No. 2019-07-008 and IRB No.2020-01-053). In addition, the need for informed consent was waived due to the use of de-identified data.

### 2.2. Comparison of Vitamin D-Related SNPs among the Global Population and East Asia

The most commonly used cut-points for serum [25(OH)D_3_] levels in adults are 11–19 ng/mL, and ≤10 ng/mL for, deficiency, and severe deficiency, respectively. However, we used the average vitamin D concentrations for each cohort instead of the prevalence of vitamin D deficiency for two reasons: first, the data among African and South Asians are limited [9]; second, the prevalence of vitamin D deficiency and average 25(OH)D_3_ concentration are related. We searched the NHGRI-EBI GWAS catalog (https://www.ebi.ac.uk/gwas/home, 30 December 2020) for SNPs associated with vitamin D measurements (EFO 0004631). The catalog included 13 studies and 546 associations. After eliminating repetitive SNPs and removing data not found in the 1000 Genome Projects database, 320 SNPs from the GWAS catalog were used for the analysis of allele frequencies associated with vitamin D concentrations.

The details and advantages of our method have been described elsewhere [19,20,21]. In brief, the population-level allele frequencies of SNPs were derived from the 1000 Genomes Project phase 3 and the Korean Reference Genome Database (KRGDB) produced by the Korea National Institute of Health in 2016. The former surveyed genetic variations among 2504 individuals from 26 worldwide populations grouped into African, East Asian, European, South Asian, and American categories based on their geographical locations and ancestry [22]. These data were downloaded (ftp://ftp.1000genomes.ebi.ac.uk/vol1/ftp/release/20130502/, last accessed: 15 January 2020). The variant coordinates were based on the human genome assembly GRCh37. The latter included data on 1722 individuals from the Korean population since the 1000 Genomes Project did not have this information [23]. Data on the population frequencies of the SNPs were downloaded from the web-based database (http://152.99.75.168:9090/KRGDB/menuPages/download.jsp/, last accessed: 15 January 2020). In order to compare the distributions of risk alleles in the Korean population, individual genotyping results from the second phase of KRGDB were obtained from 1099 individuals from the National Human Resource Bank of Korea. After statistical analysis, we performed expression quantitative trait locus (eQTL) analysis for significant SNPs using the Genotype-Tissue Expression (GTEx) portal (https://www.gtexportal.org/ accessed on 30 December 2020) for the significance of vitamin D SNP enrichment. Gene and transcript expression on the GTEx portal are shown in the Transcripts Per Million (TPM) unit, calculated as
$$Transcripts\;Per\;Million=\frac{n_{t}}{I_{t}}{\left(\underset{k\in T}{\sum }\frac{n_{k}}{I_{k}}\right)}^{-1}\times 10^{6}$$
where “*n_t_*” refers to the number of reads for transcript/gene, the normalized transcript/gene length, and “*T*” is the set of all transcripts or genes depending on whether the quantification is at the gene level. The normalized expression (norm expression) values were calculated with edgeR (https://gtexportal.org/home/documentationPage-#staticTextAnalysisMethods accessed on 30 December 2020). 

### 2.3. Calculation of Genetic Risk Scores Using SNPs Related to Vitamin D Concentration

To compare the composite genetic risk of vitamin D deficiency, we adopted the following equation provided by Mao et al. [21]:(1)$$Genetic\;risk\;score=\frac{\sum_{i=1}^{I}Xi}{2I}$$
where “*I*” refers to the number of vitamin D concentration-related SNPs, and “*Xi*” to the copies of risk alleles (*Xi* ϵ {0,1,2} the *i*th SNP. Thus, if a person had two copies of the risk allele at each vitamin D concentration-related SNP, their risk score was set as 1. In contrast, if a person had no copies, their risk score was 0. A person with a composite genetic risk score (GRS) of 1 has the highest possible genetic risk of higher vitamin D concentration, whereas a person with a score of 0 has the lowest. If copies of effect alleles (0/1/2) were randomly assigned to each SNP, the expected value of the risk score was set at 0.5. SNPs with frequency differences of more than 10% between the total (*n* = 1722) and second-phase (*n* = 1099) data of KRGDB were excluded from the GRS calculation. We used the average composite GRS to determine correlations with population vitamin D concentration data from similar geographical latitudes (51°) and fitting curve vitamin D concentration for its original endogenous population vs. GRS [9]. In addition, the correlation analysis of the difference of vitamin D concentrations was performed included both latitude factor and GRS factor since a previous study showed the impact of these on vitamin D concentration in patients of African and European ancestry [24].

### 2.4. Statistical Analyses

We used Fisher’s exact test to assess whether the effect allele at a given SNP was significantly higher or lower compared to the global population frequency in the 1000 Genomes Project database, and the *p* values were initially log_10_-transformed. In the heatmap generated to visualize allele patterns in different populations, red and blue colors were used to indicate higher and lower frequencies, respectively, compared to the global average. If the effect allele was enriched in a population, then the negative log_10_ of the *p*-value (a positive number) was used to represent the SNP associated with that population in the heatmap. In contrast, if it was depleted, then the log_10_ of the *p*-value (a negative number) was used. Statistical analyses were performed using R software version 4.0.1 (R Foundation, Vienna, Austria), and statistical significance was set at *p <* 0.05.

## 3. Results

### 3.1. Vitamin D Concentration-Related SNPs in the Global Population

We collected 320 vitamin D concentration-associated SNPs from 13 GWASs using the NHGRI-EBI catalog. We determined the effect allele frequencies (EAFs) for each of the continental groups and the Korean population based on the information from the 1000 Genomes Project and KRGDB (Appendix A). The heatmap shows how significantly the effect allele was enriched or depleted across these populations (Appendix A). In the Korean population, 106 vitamin D-related SNPs were significantly enriched, 120 were depleted, and 94 were comparable to the global EAF. The hierarchical clustering tree showed the differences among the populations, with Europeans, Americans, and South Asians in one cluster and Africans, East Asians, and Koreans in another. In addition, SNPs with significantly different frequencies among the Korean population (Log-adjusted *p*-value of Fisher’s exact test in Koreans >100 or <−100) are summarized in Table 1 and Figure 1.

From the data, rs10818769 and rs9409266 were found to be depleted in Koreans, East Asians, and Africans but enriched in Europeans. The SNP (rs10818769, rs9409266) is located in an intronic region of the *RABGAP1* gene, which encodes guanosine triphosphatase-activating protein of RAB6A, and has alleles of C > G and G > A, respectively. The major allele was detected in 85% of Europeans and 26% of Koreans. Although the *RABGAP1* gene is known to be related to body height and birth weight, these SNPs may be related to modulation of the evolution-related gene for the ethnic component of vitamin D concentration. The box plots of eQTL of the *RABGAP1* genes related to vitamin D in skin tissues of both sun-exposed and non-exposed areas show a significantly different expression, according to the alleles of rs10818769 and rs9409266 in the GTEx data (Figure 2). Comparison of allele frequency of major vitamin D-related genes, such as *GC*, *NADSYN1/DHCR7*, *CYP2R1*, and *CYP24A1*, are summarized in Appendix A.

### 3.2. Genetic Risk Scores Calculated Using SNPs Related to Vitamin D Levels

We calculated the composite GRS based on the number of copies of effect alleles at the 320 vitamin D-associated SNPs, assuming that allelic associations from most GWAS-identified variants could be replicated in non-European populations. The GRS of vitamin D concentration was highest among Europeans, followed by Americans, South Asians, East Asians, and Africans (Figure 3).

A strong correlation was observed between the vitamin D concentration from several studies [25,26,27,28,29,30] and GRS with a similar geographic latitude (51°, *R*^2^ = 0.59) in the grey dashed line (Figure 4). In addition, the vitamin D concentration for its original endogenous population vs. GRS fitted to the U curves of the black line (Figure 4). Correlation plot of the difference between vitamin D concentration and GRS using related SNPs or latitude Vitamin D concentration was strongly correlated with average GRS rather than latitude effect, when reviewing the vitamin D difference vs. GRS with *R*^2^ value of 0.9996 (A), instead of latitude difference with an *R*^2^ value of 0.6438 (B) in Figure 5.

Vitamin D concentration was strongly correlated with average genetic risk score rather than latitude effect when reviewing the vitamin D difference versus genetic risk score with an *R*^2^ value of 0.9996, instead of latitude difference with an *R*^2^ value of 0.6438.

## 4. Discussion

Vitamin D deficiency is associated with unfavorable bone conditions and chronic diseases such as cancer and diabetes [31]. Thus, in this study, we aimed to assess the different risk alleles of ethnic groups that may reflect vitamin D concentrations. We found that allele frequencies were found to differ dependent on ethnic group, and the SNP-based genetic score was shown to have a strong correlation with real-world data of vitamin D levels.

Previously conducted GWASs revealed several significant loci, including *GC*, *NADSYN1*/*DHCR7*, *CYP2R1*, and *CYP24A1* [11,12,13,14,15,16,17], that played an important role in vitamin D concentrations. Subsequently, using these significant loci (46 SNPs), Jones et al. showed variations of vitamin D levels among European, East Asian, and African populations by UVB exposure and ancestry [24]. Our study hypothesized that different allele frequencies of ethnic groups and Koreans might have significant loci for the evolution of different vitamin D concentrations regardless of environmental factors. In our study, the rs200641845 and rs7041 related *GC*, encoding vitamin D binding carrier protein, were highly depleted in Koreans and Africans. The rs3829251 and rs11233933 associated with *NADSYN1* were depleted in Africans whereas they were enriched in Koreans. This could be one piece of the evidence in relation to Koreans and Africans having a different mechanism related to low vitamin D levels, as NADSYN1/DHCR7 is involved in UVB-induced vitamin D metabolism in the skin.

Additionally, we found some SNPs [rs10818769 (*RABGAP1*), rs9409266 (*RABGAP1*), rs12881545 (*DLK1*), rs10070734 (*LINC00461*), and rs17765311 (*AC007950.2*)] that were highly underexpressed in East Asians (including Koreans) and Africans, while they wereover-expressed in Europeans. The eQTL analysis showed that rs10818769 and rs9409266 affected *RABGAP1* expression in the skin regardless of sun exposure. The pigmentation-associated allele evolution has been shown to include *SLC24A5* [32] and *RABGAP1* in a previous study [33], and *RABGAP1* was the signature gene for vitamin D deficiency and skin pigmentation. It was found to be underexpressed among East Asians (including Koreans) and Africans, while highly expressed among Europeans in our study. This gene may provide a possible link between skin pigmentation and vitamin D concentration; however, further experimental studies are needed to confirm this. This result is consistent with the nutrigenomics of vitamin D in that the main evolutionary driver of decreased skin pigmentation was the need for sufficient endogenous vitamin D production [34]. Skin color and genetic variation may explain vitamin D deficiency and adaptation to life in the latitudes [35].

The GRS was the highest among Europeans, followed by Americans, South Asians, East Asians, and Africans, and was correlated with vitamin D concentrations. This result is consistent with the estimates of 25(OH)D_3_ levels <20 ng/mL that have been reported as 24% in the USA, 37% in Canada, and 40% in Europe [36,37]. European Caucasians have been shown to have lower rates of vitamin D deficiency compared with nonwhite individuals [36,38]. In addition to genetics, environmental factors such as nutrition and sunlight exposure are important determinants of vitamin D concentration, and latitude was one of the factors considered in this study. Moreover, vitamin D deficiency is common in non-western immigrants due to low sunshine exposure, pigment skin, and low calcium intake [39]. In this regard, the comparison of multi-ethnic group data in a single country would be desirable. According to the study by Van der Meer et al., the mean vitamin D concentration was 26.8 ng/mL among the Dutch and 13.2 ng/mL among Africans in the Netherlands at a latitude of 51° [40]. These results are consistent with those of another study showing that African Americans had lower levels of vitamin D than European Americans [41]. The pooled prevalence of low vitamin D status in Africa was 33.22% (26.22–43.68%) with a cutoff of serum 25(OH)D_3_ concentration of <20 ng/mL and an overall mean of 26.8 ng/mL [42]. Furthermore, vitamin D concentrations were strongly correlated with GRS rather than latitude effects when examining GRS vs. vitamin D concentration differences instead of latitudinal differences. Thus, latitude factors should be considered for vitamin D concentration assessment in genetic models, as was performed in our study.

A major strength of our study was the inclusion of the Korean whole-genome dataset of 1722 individuals that reflected the allele frequency of SNPs related to vitamin D deficiency. Moreover, we computed the risk model using a significant number of alleles (*n* = 320) related to vitamin D compared to a previous study that used only the major loci [24]. Additionally, we did not systematically organize the new vitamin D cohort and analyze the effect; instead, we compared the data from the 1000 Genomes Project with the vitamin D-related SNP data from the GWAS catalog. Despite these strengths, there are some limitations to this study. First, the GWAS catalog contained data where the risk allele was not clearly defined according to the minor allele frequency (MAF). We did not exclude these from our study because the majority of MAFs were likely to be risk alleles. Therefore, inaccurate subgroup analysis could have arisen. To address this issue, risk allele curation is necessary for the GWAS catalog based on the results of additional large population studies using cohorts in whom vitamin D was measured. Moreover, the statistical significance of EAF in the Korean population was high and should be interpreted with caution since Fisher’s test can decrease the *p*-value as the number of subjects increases, even with the same odds ratios. Third, latitude and genomic modeling were only used for vitamin D analysis; other environmental factors (such as nutrition) were not considered. A previous study on a multi-ethnic population of Norway has shown that there are many modifiable risk factors related to 25(OH)D_3_ levels [43]. Finally, we used the composite GRS instead of polygenic risk score for two reasons: the weighted-odd ratios of vitamin D concentrations varied according to the ethnic group even for the same SNP, and as there were inaccuracies of weighted-odd ratios due to insufficient study data among the African and American populations. In the future, a polygenic risk score with the effect size-weighted odd ratio should be evaluated.

## 5. Conclusions

Our study found a substantial population difference in terms of allele frequencies in vitamin D-related SNPs. The GRS for vitamin D concentrations was higher in Europeans compared to that found in East Asians and Africans, which were highly correlated with actual data. In addition, vitamin D concentration was strongly correlated with average GRS rather than latitude effect. From the public health perspective of vitamin D deficiency, genetic variants associated with vitamin D, as well as environmental factors (latitude, UVB exposures), should be considered. Further studies are needed to identify variant SNPs in genes such as *RABGAP1* (rs10818769, rs9409266) that reflect vitamin D deficiency in East Asians or Africans and to assess their modifiable roles for evolutionary differences.

## Figures and Tables

**Figure 1 genes-12-01530-f001:**
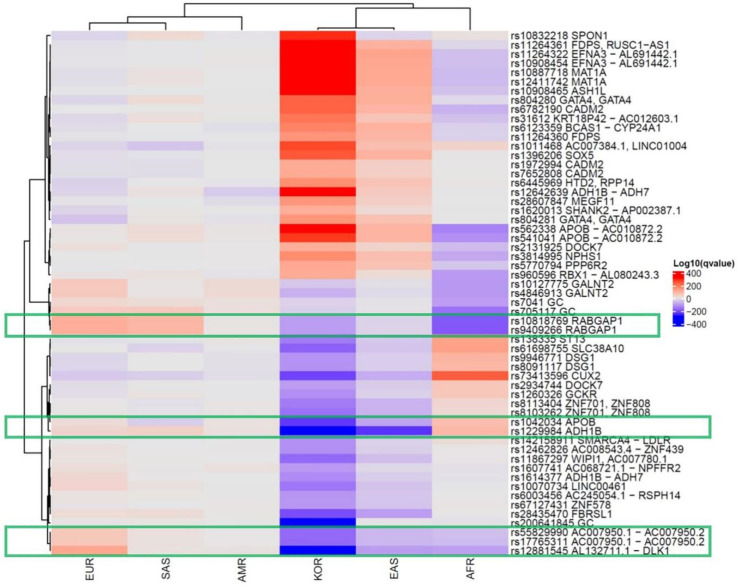
Significant single-nucleotide polymorphisms related to the vitamin D concentration in the global population. Each row shows an SNP, and each column is a population group. The red color indicates enrichment of the effect allele, whereas the purple color indicates the depletion. The following SNPs ((rs10818769 and rs9409266 (*RABGAP1*) in the red box, followed by rs12881545 (*DLK1*), rs10070734 (*LINC00461*), and rs17765311 (*AC007950.2*)) were highly depleted in East Asians (including Koreans) and Africans and were enriched in Europeans. (AMR: American, EUR: European, SAS: South Asian, AFR: African, EAS: East Asian, KOR: Korean.).

**Figure 2 genes-12-01530-f002:**
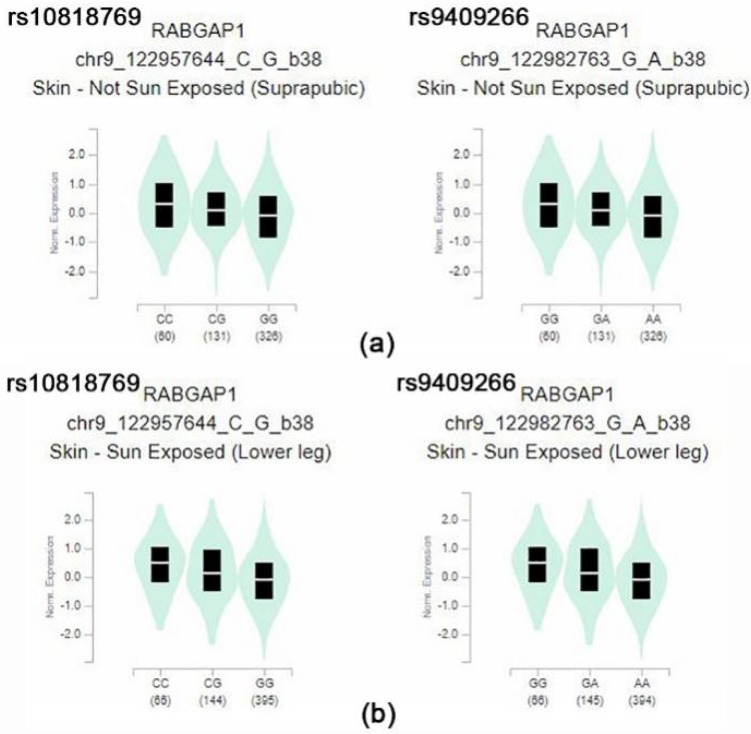
(**a**) Box plot of eQTL variant results (*p* = 3.4 × 10^−12^, *p* = 3.4 × 10^−12^, respectively): rs10818769-skin (suprapubic), rs9409266-skin (suprapubic) (**b**) box plot of eQTL variant results (*p* = 6.4 × 10^−12^, *p* = 6.0 × 10^−12^, respectively): rs10818769-skin (lower leg), rs9409266-skin (lower leg) These variants showed significantly different expression eQTL in their minor allele of the *RABGAP1*. Normalized expression (norm expression) was calculated in Transcript Per Million (TPM) units.

**Figure 3 genes-12-01530-f003:**
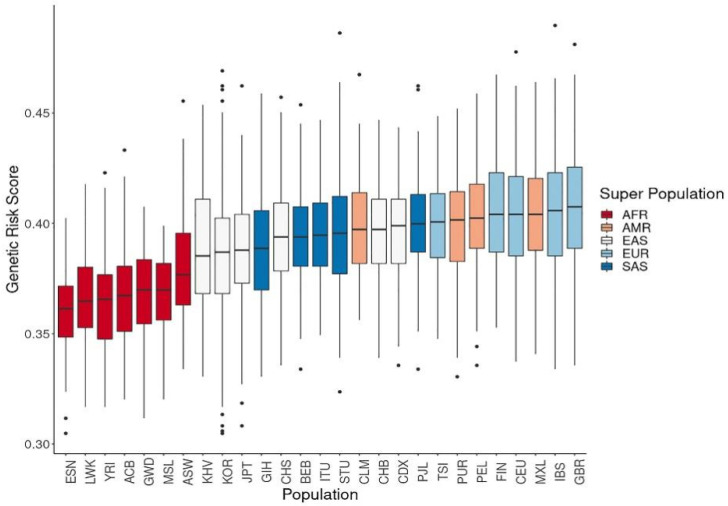
Genetic risk score calculations of vitamin D concentration using related single-nucleotide polymorphisms. The genetic risk scores calculated from allele frequencies for vitamin D concentration were highest in Europeans, followed by East Asians and Africans. (ACB: African Caribbean in Barbados; ASW: African ancestry in the Southwest USA; BEB: Bengali in Bangladesh; CDX: Chinese Dai in Xishuangbanna; CEU: Utah residents with Northern and Western European ancestry; CHB: Han Chinese in Beijing, China; CHS: Southern Han Chinese, China; CLM: Colombian in Medellin, Colombia; ESN: Esan in Nigeria; FIN: Finnish in Finland; GBR: British in England and Scotland; GIH: Gujarati Indian in Houston, TX, USA; GWD: Gambian in Western Division, Gambia; IBS: Iberian populations in Spain; ITU: Indian Telugu in the UK; JPT: Japanese in Tokyo, Japan; KOR: Korean in the Republic of Korea; KHV: Kinh in Ho Chi Minh City, Vietnam; LWK: Luhya in Webuye, Kenya; MSL: Mende in Sierra Leone; MXL: Mexican ancestry in Los Angeles, CA, USA; PEL: Peruvian in Lima, Peru; PJL: Punjabi in Lahore, Pakistan; PUR: Puerto Rican in Puerto Rico; STU: Sri Lankan Tamil in the UK; TSI: Toscani in Italy; YRI: Yoruba in Ibadan, Nigeria.).

**Figure 4 genes-12-01530-f004:**
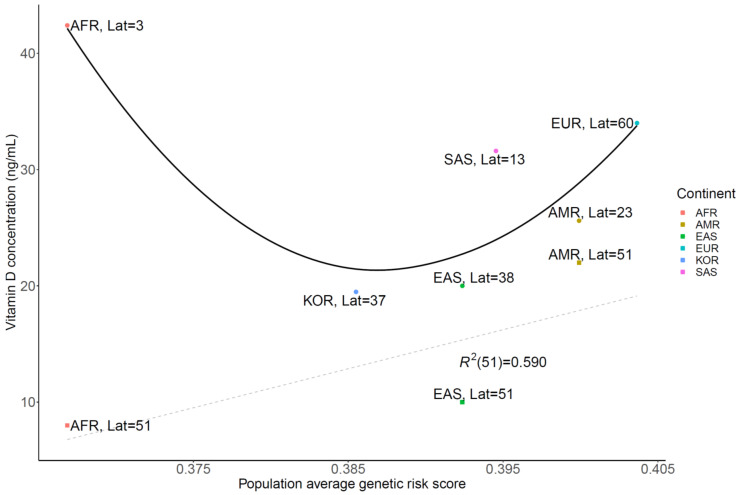
Correlation plot of vitamin D concentration and genetic risk score using related single-nucleotide polymorphisms, with latitude. Vitamin D concentration was strongly correlated with average genetic risk score at the same latitude (*R*^2^ = 0.590) in the grey dashed line. Its original endogenous population vitamin D concentration vs genetic risk score fitted in the black line. (Lat: Latitude, AMR: American, EUR: European, SAS: South Asian, AFR: African, EAS: East Asian).

**Figure 5 genes-12-01530-f005:**
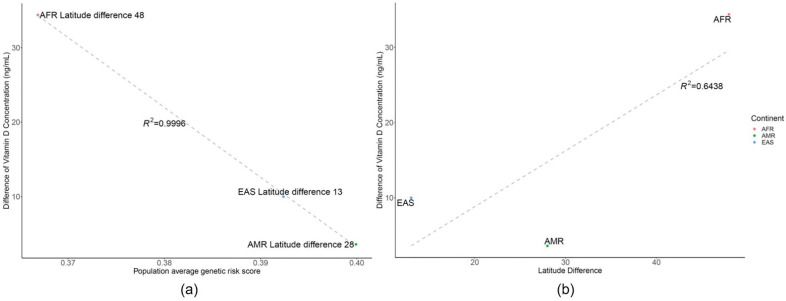
Correlation plot of the difference between vitamin D concentration and genetic risk score using related single-nucleotide polymorphisms or latitude. (**a**) The difference between vitamin D and genetic risk score, showing that a lower genetic risk score cannot compensate or generate vitamin D when this population is relocated to a higher latitude, where the regression *R*^2^ is 0.9996. (**b**) Difference between vitamin D and latitude, showing that latitude difference does not impact the difference of vitamin D as much as genetic risk score, with an *R*^2^ value of 0.6538. (AFR: African, AMR: American, EAS: East Asian).

**Table 1 genes-12-01530-t001:** Effect allele frequencies of vitamin D concentration-related single-nucleotide polymorphisms among continental groups.

SNP ID	Chr ^a^	Position	MAPPED_GENE	Function	Ref Allele ^b^	Alt Allele ^c^	Global EAF ^d^	AMR ^e^ EAF	AMR log10 P	AFR ^f^ EAF	AFR log10 P	EAS ^g^ EAF	EAS log10 P	SAS ^h^ EAF	SAS log10 P	EUR ^i^ EAF	EUR log10 P	KOR ^k^ EAF	KOR log10 P
rs2131925	chr1	63025942	DOCK7	intron_variant	G	T	0.56	0.61	1.61	0.31	−58.18	0.77	35.67	0.53	−0.97	0.69	13.25	0.785	102.29
rs10908454	chr1	155066416	EFNA3-AL691442.1	TF_binding_site_variant	G	A	0.55	0.64	4.64	0.31	−53.75	0.91	116.71	0.51	−1.48	0.48	−3.99	0.919	312.61
rs11264322	chr1	155087933	EFNA3-AL691442.1	intergenic_variant	G	A	0.54	0.61	2.91	0.3	−53.97	0.91	122.04	0.5	−1.51	0.45	−6.28	0.919	312.61
rs11264360	chr1	155284586	FDPS	intron_variant	T	A	0.35	0.3	−1.79	0.2	−25.83	0.7	91.61	0.3	−2.46	0.27	−5.79	0.638	149.83
rs11264361	chr1	155289545	FDPS, RUSC1-AS1	non_coding_transcript_exon_variant	T	G	0.35	0.3	−1.79	0.21	−22.25	0.71	96.83	0.31	−1.66	0.26	−7.22	0.801	312.61
rs10908465	chr1	155389688	ASH1L	intron_variant	C	T	0.34	0.31	−0.75	0.14	−48.92	0.71	102.27	0.31	−1.04	0.28	−3.40	0.811	312.61
rs562338	chr2	21288321	APOB-AC010872.2	intergenic_variant	A	G	0.73	0.82	6.07	0.35	−138.10	0.98	91.24	0.87	21.07	0.8	5.29	0.995	311.61
rs541041	chr2	21294975	APOB-AC010872.2	intergenic_variant	G	A	0.75	0.82	3.96	0.4	−119.94	0.99	93.79	0.88	19.24	0.8	3.04	0.995	282.11
rs6782190	chr3	85639672	CADM2	intron_variant	G	A	0.65	0.73	4.16	0.37	−73.11	0.94	91.71	0.64	−0.23	0.65	0.00	0.936	231.64
rs200641845	chr4	72620895	GC	intron_variant	T	A	0.42	0.43	0.14	0.34	−6.80	0.45	1.03	0.47	2.24	0.45	1.02	0.080	−285.57
rs1607741	chr4	72719033	AC068721.1-NPFFR2	intergenic_variant	G	C	0.58	0.72	11.29	0.49	−8.10	0.42	−19.40	0.61	0.98	0.73	18.02	0.338	−106.15
rs1614377	chr4	100279332	ADH1B-ADH7	intergenic_variant	G	A	0.15	0.17	0.61	0.095	−6.69	0.026	−33.64	0.18	1.55	0.31	28.74	0.012	−125.88
rs12642639	chr4	100301241	ADH1B-ADH7	intergenic_variant	C	A	0.38	0.14	−37.19	0.34	−2.07	0.65	54.04	0.51	12.56	0.21	−24.90	0.825	312.61
rs10070734	chr5	87940026	LINC00461	intron_variant	T	C	0.5	0.46	−1.12	0.46	−1.92	0.25	−48.15	0.61	9.11	0.71	33.25	0.217	−155.96
rs31612	chr5	108996643	KRT18P42-AC012603.1	intergenic_variant	T	C	0.32	0.28	−1.27	0.12	−51.58	0.61	63.57	0.46	15.17	0.18	−18.77	0.654	201.59
rs804280	chr8	11612698	GATA4, GATA4	intron_variant	C	A	0.73	0.69	−1.31	0.6	−18.48	0.99	103.54	0.87	21.07	0.56	−24.11	0.975	231.56
rs10818769	chr9	125719923	RABGAP1	intron_variant	C	G	0.49	0.58	4.58	0.071	−197.61	0.31	−24.91	0.83	89.25	0.85	104.26	0.260	−100.85
rs9409266	chr9	125745042	RABGAP1	intron_variant	G	A	0.49	0.58	4.58	0.07	−198.46	0.31	−24.91	0.83	89.25	0.85	104.26	0.260	−101.09
rs10887718	chr10	82042624	MAT1A	intron_variant	C	T	0.66	0.67	0.16	0.41	−58.77	0.97	114.96	0.76	8.79	0.54	−11.52	0.979	312.61
rs12411742	chr10	82042782	MAT1A	intron_variant	G	A	0.66	0.67	0.16	0.41	−58.77	0.97	114.96	0.76	8.79	0.54	−11.52	0.979	312.61
rs1620013	chr11	71089210	SHANK2-AP002387.1	intergenic_variant	C	T	0.48	0.45	−0.72	0.45	−1.21	0.66	24.42	0.53	2.16	0.3	−24.67	0.717	104.75
rs1396206	chr12	24576859	SOX5	intron_variant	A	T	0.7	0.67	−0.78	0.67	−1.37	0.97	94.88	0.61	−6.85	0.58	−12.07	0.967	245.87
rs12881545	chr14	101176212	AL132711.1-DLK1	TF_binding_site_variant	G	C	0.26	0.38	9.25	0.029	−97.30	0.0069	−103.17	0.35	7.30	0.65	116.98	0.000	−286.57
rs17765311	chr15	63789952	AC007950.1-AC007950.2	regulatory_region_variant	A	C	0.16	0.2	1.80	0.016	−58.56	0.001	−65.52	0.26	11.55	0.4	57.06	0.003	−174.20
rs55829990	chr15	63790642	AC007950.1-AC007950.2	intergenic_variant	T	C	0.16	0.2	1.80	0.019	−54.99	0.002	−63.59	0.26	11.55	0.4	57.06	0.003	−174.20
rs28607847	chr15	66284913	MEGF11	intron_variant	G	A	0.38	0.22	−15.36	0.36	−0.69	0.62	42.80	0.38	0.00	0.29	−6.89	0.668	149.95
rs3814995	chr19	36342212	NPHS1	missense_variant	C	T	0.29	0.34	1.85	0.056	−84.87	0.6	74.07	0.24	−2.58	0.31	0.62	0.587	162.75
rs6123359	chr20	52714706	BCAS1-CYP24A1	regulatory_region_variant	A	G	0.21	0.1	−11.63	0.083	−28.70	0.55	96.31	0.22	0.27	0.11	−13.14	0.511	180.25
rs960596	chr22	41393520	RBX1-AL080243.3	intergenic_variant	C	T	0.27	0.2	−3.71	0.026	−107.11	0.43	21.65	0.38	10.42	0.35	5.99	0.508	108.51

*p*-value: adjusted Fischer’s test, statistical significance was set at *p* < 0.05. a: chromosome. b: reference allele. c: alternative allele. d: effect allele frequency. e: Americans. f: Africans. g: East Asians. h: South Asians. i: Europeans. k: Koreans.

## Data Availability

The raw datasets generated and analyzed during the current study are not publicly available since any data providing the whole-genome sequencing data is considered to be personal property by the Korea Bioethics law. However, the raw whole-genome sequencing data for research are available at the reasonable request under the permission of the National Biobank of Korea contact at [http://nih.go.kr/biobank/cmm/main/mainPage.do?/, accessed on 15 January 2020] and e-mail [biobank@korea.kr]. The allele frequency of Korea reference genome data base (KRGDB) is available [http://152.99.75.168:9090/KRGDBDN/dnKRGinput.jsp, accessed on 15 January 2020], files required are all three of ‘the totally merged sets’ of common variants, rare variants, and indels. The 1000genomes data is available, all the files from the following folder were downloaded, [ftp://ftp.1000genomes.ebi.ac.uk/vol1/ftp/release/20130502/] (last accessed: 15 January 2020). The genome-wide association study (GWAS) catalog data is available in the (NHGRI-EBI, [https://www.ebi.ac.uk/gwas/docs/file-downloads, accessed on 15 January 2020], “All associations v1.0.2—with added ontology annotations, GWAS Catalog study accession numbers and genotyping technology”, December 2020).

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
