# Peer review of "Risk Allele Frequency Analysis of Single-Nucleotide Polymorphisms for Vitamin D Concentrations in Different Ethnic Group"

_genes, 2021, doi:10.3390/genes12101530_

Round 1
Reviewer 1 Report
In this manuscript, Yoon et al. studied 320 single-nucleotide polymorphisms (SNPs) associated with vitamin D concentrations, with population-level allele frequencies data from 1KG Project and Korean Reference Genome Database. In their results, Yoon et al. identified depleted SNPs (rs200641845 and rs7041) and enriched SNPs (rs6123359 and rs3829251) in Koreans, also in East Asians and Africans. Additionally, in genes including RABGAP1, DLK1, et al., they found SNPs underexpressed in these populations in contrast to Europeans, provided a possible link between skin pigmentation and vitamin D concentration. Finally, they calculated the SNP-based genetic risk score (GRS) and performed correlation analysis with vitamin D concentration in different populations. The difference in vitamin D concentration was highly correlated with genetic factors rather than latitude effects.
This is a very interesting work. The authors demonstrated clear results and provided a thorough discussion. However, several minor issues need to be addressed by the authors.
Minor issues:
- In Figure 1, the red boxes blocked some text. Please replace it by mark the text with different color or use other proper methods to illustrate it. In addition, the title of the color legend (Log_qvalue) should be "log10(qvalue)"
- In Line 162, it should be "population group", instead of continent.
- In Figure 2, the authors should describe the methods related to the boxplot. For example, what is "Norm Expression". Moreover, this is not a very good way to illustrate the significant difference between different genotypes. What if using other widely used normalization methods like TPM or FPKM.
- In Figure 4, the numbers after the population IDs look like latitudes. The author should indicate this in the legend.
- In Figure 5, it should be R2, not R2.
- After these figures, there are some figure legend paragraphs were formatted as manuscript text. It's confusing to readers.
Author Response
Comments and Suggestions for Authors
In this manuscript, Yoon et al. studied 320 single-nucleotide polymorphisms (SNPs) associated with vitamin D concentrations, with population-level allele frequencies data from 1KG Project and Korean Reference Genome Database. In their results, Yoon et al. identified depleted SNPs (rs200641845 and rs7041) and enriched SNPs (rs6123359 and rs3829251) in Koreans, also in East Asians and Africans. Additionally, in genes including RABGAP1, DLK1, et al., they found SNPs underexpressed in these populations in contrast to Europeans, provided a possible link between skin pigmentation and vitamin D concentration. Finally, they calculated the SNP-based genetic risk score (GRS) and performed correlation analysis with vitamin D concentration in different populations. The difference in vitamin D concentration was highly correlated with genetic factors rather than latitude effects.
This is a very interesting work. The authors demonstrated clear results and provided a thorough discussion. However, several minor issues need to be addressed by the authors.
→ We appreciate the feedback. We have rewritten the manuscript according to all the reviewer suggestions. Our answers to each comment are provided in a point-by-point manner in blue lighted.
Minor issues:
- In Figure 1, the red boxes blocked some text. Please replace it by mark the text with different color or use other proper methods to illustrate it. In addition, the title of the color legend (Log_qvalue) should be "log10(qvalue)"
→ We appreciate the feedback. We have made the correction as suggested (Figure 1).
- In Line 162, it should be "population group", instead of continent.
→ We appreciate the feedback. We have made the correction as suggested (line 172).
- In Figure 2, the authors should describe the methods related to the boxplot. For example, what is "Norm Expression". Moreover, this is not a very good way to illustrate the significant difference between different genotypes. What if using other widely used normalization methods like TPM or FPKM.
→ We appreciate the feedback. This comment is an important question in expression analysis. The GTEx portal (https://www.gtexportal.org/home/faq#tpm) provided TPM units rather than the RPKM unit. Hence, we used “Normalized (norm) Expression” in the TPM unit. We changed of figures and added this comment in detail (line 106-113 and Figure 2 legend).
- In Figure 4, the numbers after the population IDs look like latitudes. The author should indicate this in the legend.
→ Thank you for pointing this out. We have changed accordingly and added abbreviations in Figure 4 legend.
- In Figure 5, it should be R2, not R2.
→ We appreciate the feedback. We have made the correction as suggested (Figure 5).
- After these figures, there are some figure legend paragraphs were formatted as manuscript text. It's confusing to readers.
→ We agree that it would confuse readers. We have changed to match the figure legend and manuscript area.

Reviewer 2 Report
Vitamin D deficiency is currently an increasing problem worldwide. The authors decided to look for the cause of vitamin D level alterations in the single nucleotide polymorphisms in genes related to vitamin D metabolism in healthy subjects of different ethnicity. In my opinion, the authors should add more detailed explanation why they decided to analyze those genes, maybe 2-3 sentences about vitamin D metabolism?
Vitamin D in 25-hydroxylated form should be written 25(OH)D3.
Lines 43-45 “In addition, the deficiency status differed even at similar latitudes, as it ranged from 18.6% (26.0 ± 7.04 ng/mL) among Norwegians and was 60.3% (18.0 g/mL) among the Finnish.” - there is inconsistency in tenses, the sentence is unclear.
Lines 206-216: The short forms used by authors in the figure 3 should be written below the legend not in the text.
Lines 239- 243: The legend should be below the figure 4.
Lines 258-261: “In Koreans, SNPs 258 such as rs200641845 and rs7041 (GC gene) were depleted, whereas rs6123359 (CYP24A1 259 gene) and rs3829251 (NADSYN1 gene) were enriched. These could be the alleles related to low vitamin D levels among East Asians and Africans.” - those conclusions are unclear: which allele precisely? And why the allele can be related to low vitamin D level in those ethnic groups? It should be more clarified.
Lines 313-315: “A previous study on a multi-ethnic population of Norway has shown that many modifiable risk factors related to vitamin D levels.” - the sentence is not clear.
Lines 315-318: “Finally, we used the composite GRS, which did not include 315 the weighted-size effect, as the weighted-odd ratios varied according to the ethnic group 316 even for the same SNP, and as there are inaccuracies due to insufficient study data among 317 the African and American populations.” - the sentence is too long and confusing.
Please check the punctuation and spelling in the manuscript.
Author Response
Comments and Suggestions for Authors
Vitamin D deficiency is currently an increasing problem worldwide. The authors decided to look for the cause of vitamin D level alterations in the single nucleotide polymorphisms in genes related to vitamin D metabolism in healthy subjects of different ethnicity. In my opinion, the authors should add more detailed explanation why they decided to analyze those genes, maybe 2-3 sentences about vitamin D metabolism? → Thank you for the kind comments for improving the paper. We have rewritten the manuscript according to all the reviewer suggestions. Our answers to each comment are provided in a point-by-point manner in blue lighted.
We added comments on vitamin D metabolism accordingly (lines 31-36 and lines 56-58).
Vitamin D in 25-hydroxylated form should be written 25(OH)D3. → We appreciate the feedback. We have made the correction as suggested in the text.
Lines 43-45 “In addition, the deficiency status differed even at similar latitudes, as it ranged from 18.6% (26.0 ± 7.04 ng/mL) among Norwegians and was 60.3% (18.0 g/mL) among the Finnish.” - there is inconsistency in tenses, the sentence is unclear. → We appreciate the feedback. We have made the correction as follows. “In addition, the deficiency status differed even at similar latitudes, as it ranged from 18.6% (26.0 ± 7.04 ng/mL) among Norwegians and to 60.3% (18.0 ng/mL) among the Finnish [9]. (lines 46-47).”
Lines 206-216: The short forms used by authors in figure 3 should be written below the legend not in the text. → We appreciate the feedback. We have made the correction accordingly in the figure 3 legend (lines 205-215).
Lines 239- 243: The legend should be below the figure 4. → We appreciate the feedback. We have made the correction as suggested in figure 4 legend (lines 227-229).
Lines 258-261: “In Koreans, SNPs 258 such as rs200641845 and rs7041 (GC gene) were depleted, whereas rs6123359 (CYP24A1 259 gene) and rs3829251 (NADSYN1 gene) were enriched. These could be the alleles related to low vitamin D levels among East Asians and Africans.” - those conclusions are unclear: which allele precisely? And why the allele can be related to low vitamin D level in those ethnic groups? It should be more clarified. → We appreciate the feedback. The authors agree that this expression is not clear and misleading for readers. We have made the correction as suggested (lines 253-259).
Lines 313-315: “A previous study on a multi-ethnic population of Norway has shown that many modifiable risk factors related to vitamin D levels.” - the sentence is not clear. → We appreciate the feedback. We have made the correction as suggested as follows. “A previous study on a multi-ethnic population of Norway has shown that there are many modifiable risk factors related to 25(OH)D3 levels. (lines 311-313).”
Lines 315-318: “Finally, we used the composite GRS, which did not include 315 the weighted-size effect, as the weighted-odd ratios varied according to the ethnic group 316 even for the same SNP, and as there are inaccuracies due to insufficient study data among 317 the African and American populations.” - the sentence is too long and confusing. → We appreciate the feedback. We have made the correction as follows. “Finally, we used the composite GRS instead of a polygenetic risk score for two reasons: the weighted-odd ratios of vitamin D concentrations varied according to the ethnic group even for the same SNP, and as there were inaccuracies of weighted-odd ratios due to insufficient study data among the African and American populations. (lines 313-317).”
